# Laser-induced topological spin switching in a 2D van der Waals magnet

**Maya Khela**[1], **Maciej Dąbrowski** [2] ✉, **Safe Khan**[3], **Paul S. Keatley** [2],
**Ivan Verzhbitskiy**[4,5], **Goki Eda** [4,5,6], **Robert J. Hicken** [2],
**Hidekazu Kurebayashi** [3,7,8] & **Elton J. G. Santos** [1,9] ✉

Two-dimensional (2D) van der Waals (vdW) magnets represent one of the most promising horizons for energy-efficient spintronic applications because their broad range of electronic, magnetic and topological properties. However, little is known about the interplay between light and spin properties in vdW layers. Here we show that ultrafast laser excitation can not only generate different type of spin textures in $CrGeTe_3$ vdW magnets but also induce a reversible transformation between them in a topological toggle switch mechanism. Our atomistic spin dynamics simulations and wide-field Kerr microscopy measurements show that different textures can be generated via high-intense laser pulses within the picosecond regime. The phase transformation between the different topological spin textures is obtained as additional laser pulses are applied to the system where the polarisation and final state of the spins can be controlled by external magnetic fields. Our results indicate laser-driven spin textures on 2D magnets as a pathway towards reconfigurable topological architectures at the atomistic level.

The finding of long-range magnetic ordering in 2D magnetic materials has been attracting considerable research interest on different forefronts ranging from fundamentals up to energy-efficient applications[1-11]. In this context, the ability of spin-light coupling to control the magnetic properties of atomically thin layers for integration in all-optical switching (AOS) devices[12-14] might provide a feasible pathway for faster and low-power magneto-optical implementations. Previously demonstrated on different compounds under ultrafast laser pulses[15-18], AOS has potential to give a major impact on data memory and storage technologies due to the fast, and scalable character of optical probes. Apart from the switching between various magnetic states under specific circumstances of field, laser energy and temperature, optical pulses can also lead to formation of

different spin textures with specific topology or chirality[19-21]. In particular, topologically non-trivial spin quasiparticles that can be erased, nucleated and spatially manipulated as in writing and register-shifting of logical bits[6,22-26] have been attracting substantial attention. The inherent localized nature of the spin textures and their coupling with the environment which is governed by exchange and spin-orbit interactions define the underlying femtosecond time scale of magnetic dynamics.

Different spin textures have been previously observed in vdW magnets (e.g., $Fe_3GeTe_2$, $CrCl_3$, $Cr_2Te_3$, $Fe_5GeTe_2$, $CrGeTe_3$). They range from magnetic bubbles[4,27,28] and stripe domains[29], up to topological quasiparticles, such as skyrmions[30-36] and merons[37,38], as recently summarised in the 2D Magnetic Genome[9]. These evidences

[1]Institute for Condensed Matter Physics and Complex Systems, School of Physics and Astronomy, The University of Edinburgh, EH9 3FD Edinburgh, UK. [2]Department of Physics and Astronomy, University of Exeter, EX4 4QL Exeter, UK. [3]London Centre for Nanotechnology, University College London, 17-19 Gordon Street, London WCH1 0AH, UK. [4]Department of Physics, National University of Singapore, Singapore, Singapore. [5]Centre for Advanced 2D Materials and Graphene Research Centre, Singapore, Singapore. [6] Department of Chemistry, National University of Singapore, 3 Science Drive 3, Singapore 117543, National University of Singapore, Singapore, Singapore. [7]Department of Electronic & Electrical Engineering, UCL, London WC1E 7JE, UK. [8]WPI Advanced Institute for Materials Research, Tohoku University, 2-1-1, Katahira, Sendai 980-8577, Japan. [9]Higgs Centre for Theoretical Physics, The University of Edinburgh, EH9 3FD Edinburgh, UK. ✉e-mail: M.K.Dabrowski@exeter.ac.uk; esantos@ed.ac.uk

provide a broad territory for exploration either in more fundamental levels or in functional applications where the control of spin textures in device platforms is the ultimate step. On that, the formation of skyrmions in vdW layers has been mainly due to the intrinsic magnetic features of the layers (e.g., sizeable Dzyaloshinskii-Moriya interactions, dipolar fields, etc.) or the application of external magnetic fields in order to break any inversion symmetry in the system. Nevertheless, the use of light probes such ultrafast laser pulses to generate and manipulate topological spin textures on 2D vdW compounds is currently unknown. There have been recent reports on pump probe measurements on 2D magnets[39–41], but the demonstration of the formation of spin textures via ultrafast laser excitations is yet to be reported.

Here we show that ultrashot laser excitations can be used to imprint spin textures with different chiralities on the centrosymmetric CrGeTe₃ magnet. We observe that skyrmions, anti-skyrmions, stripe domains can be generated throughout the surface after the application of laser pulses with high stability after thermal equilibration. Additional laser pulses can manipulate their topological character inducing the transformation of skyrmions into stripe domains, and vice versa, with an external magnetic field providing control on the polarity of the final state. This toggle switching mechanism is observed to be complete and reversible in CrGeTe₃ providing a practical way to write and erase information with topological features in a short time scale.

## Results

To probe the magnetic properties of CrGeTe₃ crystal we use a complementary suite of theoretical and experimental techniques comprising multiscale theoretical approximation[4,6,8,37,42–43] and wide-field Kerr microscopy (WFKM) in a polar geometry[44] (see *Methods* for details). We start showing theoretically that ultrafast laser pulses (Fig. 1a) can be used to drive the formation of different spin textures on CrGeTe₃ at different conditions. We describe the interactions via a spin Hamiltonian including most of the contributions previously found to play a role in the magnetic properties of 2D materials[42]:

$$\mathscr{H} = -\frac{1}{2}\sum_{i,j}\mathbf{S}_i\mathscr{I}_{ij}\mathbf{S}_j - \frac{1}{2}\sum_{i,j}K_{ij}(\mathbf{S}_i\cdot\mathbf{S}_j)^2 - \sum_i D_i(\mathbf{S}_i\cdot\mathbf{e})^2 - \sum_{i,j}\mathbf{A}_{ij}\cdot(\mathbf{S}_i\times\mathbf{S}_j) - \sum_i \mu_i\mathbf{S}_i\cdot\mathbf{B_{dp}}$$

(1)

where $i$, $j$ represent the atoms index, $\mathscr{I}_{ij}$ represents the exchange tensor that for CrGeTe₃ contains only the diagonal exchange terms, $\mathbf{A}_{ij}$ is the Dzyaloshinskii-Moriya interaction (DMI), $K_{ij}$ is the biquadratic exchange interaction[42], $D_i$ the uniaxial anisotropy, which is oriented out of plane ($\mathbf{e} = (0, 0, 1)$), and $\mathbf{B_{dp}}$ is the dipolar field calculated using the macrocell method[4]. The exchange interactions for CrGeTe₃ have been previously parameterized from first-principles calculations[2] and contains up to three nearest neighbors. The inclusion of biquadratic exchange[42], four-site four spin interactions and next-nearest exchange interactions have been shown to lead to the stabilisation of non-trivial spin structures in vdW magnets[37,45] and thin films[46]. For the latter, large effects on the energy barriers preventing skyrmion and antiskyrmion collapse into the ferromagnetic state have been observed. In this context, we have also investigated the role of the different interactions in Eq. (1) in stabilising topological spin textures in CrGeTe₃. Supplementary Table 1 provides a summary of the main driving forces used, where the presence of DMI is crucial (see Supplementary Movie S1) for the formation of skyrmions and more complex spin textures as discussed below. Even though the system is centrosymmetric which rules out DMI at first-nearest neighbours, the same is not applicable for second-nearest neighbours where a sizeable DMI is present. This magnitude has been both measured and calculated using inelastic neutron scattering[47] and ab initio[48] simulations, respectively, resulting in $|\mathbf{A}_{ij}| \approx 0.31$ meV. The Landau-Lifshitz-Gilbert (LLG) equation is used to describe the dynamics[4,42] at different times and

temperatures. We include the effect of the laser pulse and further relaxations of the spins and ionic lattice via the two-temperature model (2TM)[43,49] (see *Methods*). In the 2TM simulations the spin interactions (e.g., bilinear exchange, anisotropic exchange, biquadratic exchange, DMI, single-ion anisotropy) are not modified by the excitations, but rather the ultrashort pulses change the spin dynamics under different fluences, as commented below. This is an effect of the time-evolution of the system to minimize the energy towards thermal equilibration after being optically excited

Figure 1b shows that as a short laser pulse is applied into the system at 100 ps, the electron ($T_{elec}$) and phonon ($T_{phon}$) reservoir temperatures are substantially modified as time evolves. An initial sharp increase of $T_{elec}$ is observed within the sub-picosecond regime (~0.5 ps) taking the system above the Curie temperature $T_C = 68$ K[2]. $T_{phon}$ follows the variation of $T_{elec}$ as thermal relaxation takes place shortly after the excitation. The behaviour of both temperatures is dependent on the fluence energy applied to the layers displaying variation on the stabilisation of $T_{elec}$ and $T_{phon}$ towards equilibrium. That is, the higher the fluence the larger the stabilised temperatures. This affects directly how the magnetisation ($M_z/M_s$) responds to the laser pulse, which is strongly fluence-dependent (Fig. 1c). At low fluence magnitudes (e.g., 0.03 mJ cm⁻²), the demagnetisation process completes before the electron-phonon temperature equilibration is reached (0.34 K). Supplementary Table S2 shows all the equilibration temperature at the different fluences used. There is a short decay of the magnetisation from the saturation state to $M_z/M_s = 0.75$ which rapidly recovers on the timescale of ~4 ps. This demagnetisation process can be classified as type-I dynamics as a single-step demagnetization within the electron-phonon equilibration is achieved[50]. At higher fluences (i.e., 0.2 mJ cm⁻²) there is an initial rapid demagnetisation ($M_z/M_s = 0.04$) of the CrGeTe₃ (Fig. 1c) driven by the electron temperature. After the electron-phonon temperature equilibration is achieved (2.25 K) the magnetisation does not significantly recover, and instead a second, far slower stage of demagnetisation proceeds which is determined by the phonon temperature. This is a two-stage demagnetisation process defined as a type-II dynamics[50]. Indeed, this result is in good agreement with recent time-resolved magneto-optical Kerr effect measurements[39] at similar energy fluence.

Interestingly, between the low and large fluence limits, there is a transitional regime where a rapid and almost complete recovery of the magnetisation occurs within 200 ps initially following a type-I dynamics. Nevertheless, over a much longer timescale (>200 ps) the system enters in a second demagnetisation stage steered by the phonon temperature as in a type-II dynamics. In this transitional domain, the laser pulse drives the formation of a variety of magnetic objects such as bubbles (circular and elongated), spirals or stripe domains, and donut-shaped textures (e.g., 0.14 mJ cm⁻² in Fig. 1d–f). By analysing the local spin distributions of the different textures (Fig. 1g–i), and through the computation of the total topological charge $N_{sk}$ via[37,51,52]:

$$N_{sk} = \frac{1}{4\pi}\int \mathbf{n}\cdot\left(\frac{\partial\mathbf{n}}{\partial x}\times\frac{\partial\mathbf{n}}{\partial y}\right)d^2\mathbf{r}$$

(2)

where $\mathbf{n}$ is the direction vector of magnetisation $\mathbf{M}$ (e.g., $\mathbf{n} = \frac{\mathbf{M}}{|\mathbf{M}|}$), we identify the formation of skyrmions ($N_{sk} = -1$) and anti-skyrmions ($N_{sk} = +1$) of Bloch-type, and skyrmioniums ($N_{sk} = 0$)[48,53]. We estimated their diameters in the range of 20–35 nm using the magnetic parameters (e.g., exchange interactions, anisotropies, DMI)[47,48] available for CrGeTe₃. It is worth mentioning that these diameters can be changed with further optimisation of the parameters as previously demonstrated[54]. This associated with the limitations in length scale of the atomistic spin dynamics methods[55,56] resulted in smaller diameters relative to the experiments as showed below. We however assumed a more qualitative approximation to demonstrate the existence of non-trivial spin textures into the layer, and how to control them through

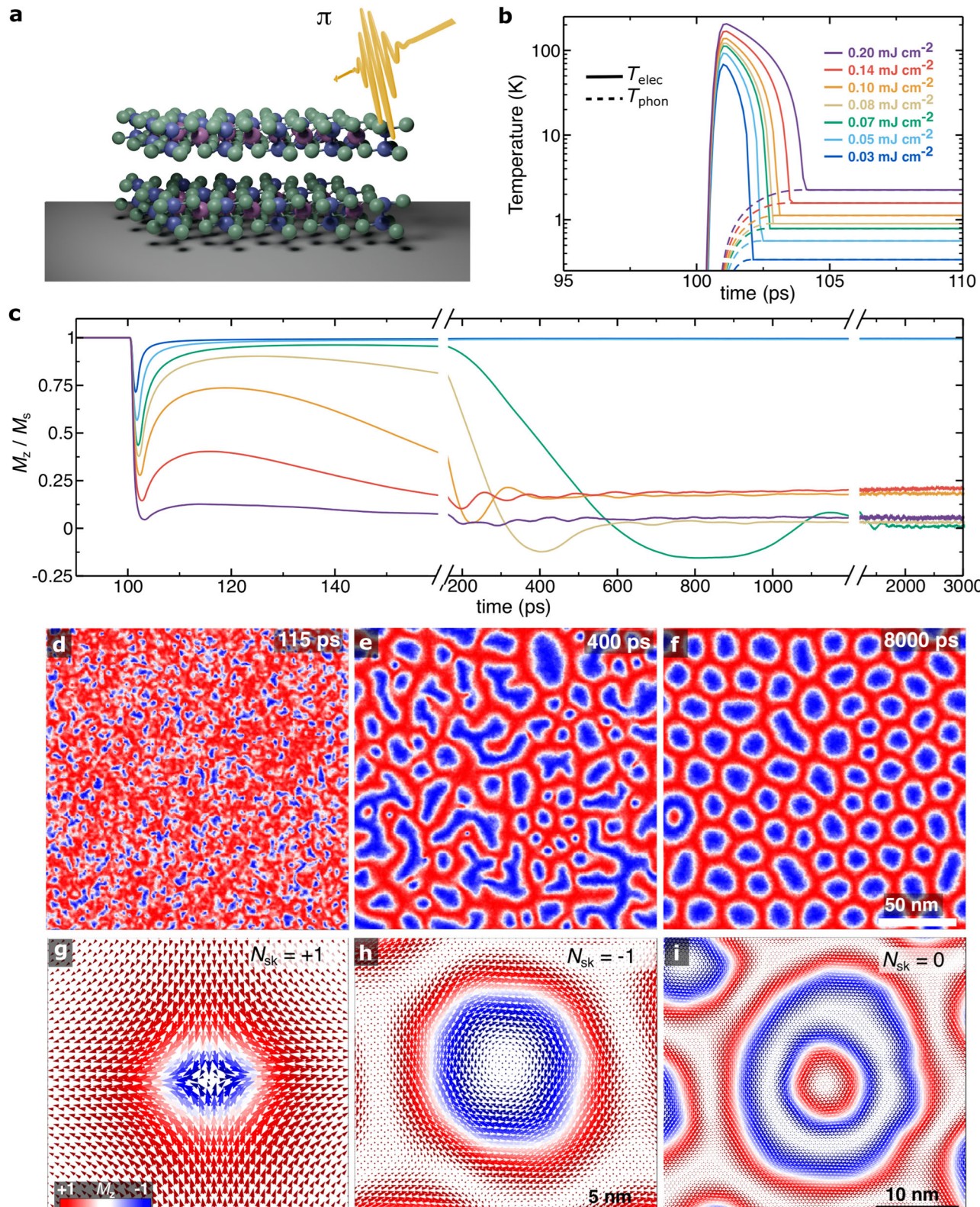

**Fig. 1 | Laser-induced demagnetisation processes and spin textures. a** Schematic of the ultrafast laser pulses with linear polarisation $\pi$ on CrGeTe$_3$. **b** Evolution of the electronic ($T_{elec}$) and phonon ($T_{phon}$) temperatures using the 2TM approach after the application of a single pulse at different energy fluences (mJ cm$^{-2}$). **c** Variation of the out-of-plane magnetisation $M_z/M_s$ versus time (ps) at the fluences showed in **b**. Line breakings are used to separate the three regimes observed in the simulations. **d**–**f** Snapshots of the spin dynamics ($M_z$) following the pulse at 115 ps, 400 ps, and 8000 ps, respectively. A fluence of 0.14 mJ cm$^{-2}$ was used. **g**–**i** Local view of the spin textures with their respective topological number $N_{sk} = +1$ (anti-skyrmion), $-1$ (skyrmion), and 0 (skyrmionium), respectively. The scale bar of 5 nm applies to both **g** and **h**. Skyrmions are generally more energetically stable than anti-skyrmions or skyrmionium even though both can stay for several nanoseconds and disappear as observed in our spin dynamics simulations.

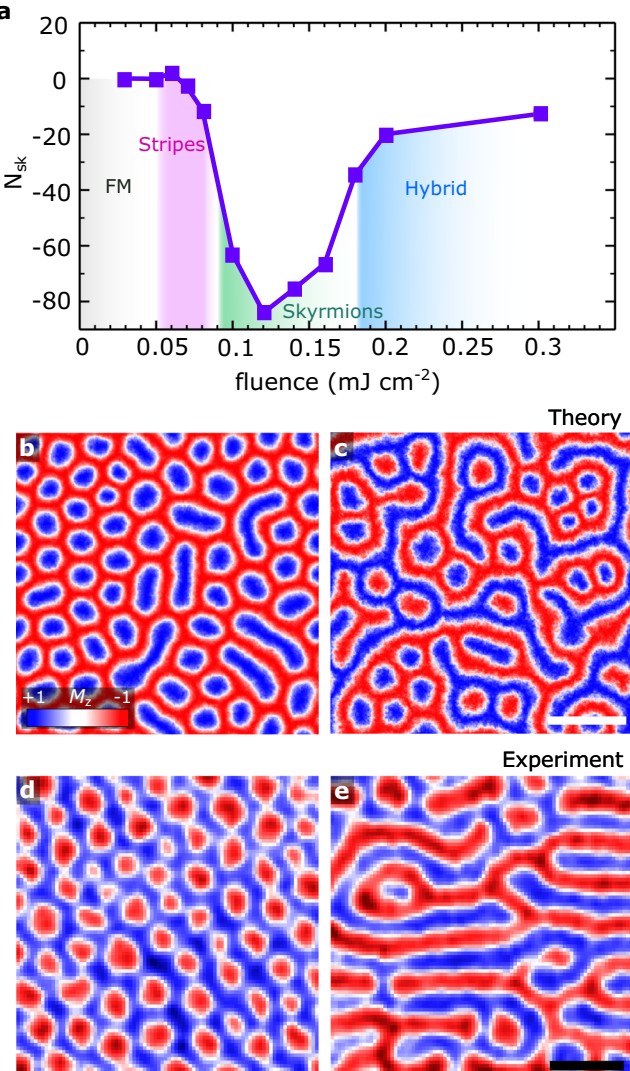

**Fig. 2 | Fluence-dependent spin textures. a** Final topological charge ($N_{sk}$) versus energy fluence (mJ cm$^{-2}$) of the applied laser. The different energies determine the spin states created in CrGeTe$_3$ within the range considered: spin polarised ferromagnetic (FM), magnetic stripes, skyrmions and hybrid (stripes and skyrmions coexisting). In each case, CrGeTe$_3$ was in a spin-polarised saturated state ($M_z = 1$) prior to the laser pulse. **b–c** Snapshots of the simulation results for 0.10 mJ cm$^{-2}$ and 0.30 mJ cm$^{-2}$, respectively. The out-of-plane magnetisation $M_z$ after 3 ns relaxation following a single laser pulse is used. The white scale bar is 50 nm. **d, e** Experimental results obtained via WFKM after 100 and 10 laser pulses, respectively. The black scale bar is 5 μm and the laser fluence is 16 mJ/cm$^2$. The measurements were performed at 18.5 K. A 25 mT magnetic field is used.

application of the laser pulse, ferromagnetic recovery occurs in localised regions of the surface which is driven by short-range exchange interactions. This localised recovery or magnon localisation, results in the nucleation of small, non-equilibrium magnetic structures known as magnon droplets[60]. The nucleation of these droplets at an early stage of the dynamics can be seen in Supplementary Fig. 1 at different fluences. Within a time scale of a few picoseconds, these droplets can split, merge and scatter until thermal equilibrium is reached which induced the formation of more stable textures with a defined spin configuration (>200 ps). Once this magnon coalescence step[60] is finished, the magnon droplets evolved into skyrmions with a specific $N$ and energetic stability against thermal fluctuations.

We observed that this phenomenon can be tuned by adjusting the energy fluence applied to the system. That is, different types of magnetic textures can be created in CrGeTe$_3$ with the laser pulse. At low fluences, 0.001–0.055 mJ cm$^{-2}$ (Fig. 2a, and Supplementary Movie S2) the final topological charge is nearly zero with the magnetisation in the type-I dynamics. This is expected at this weak spin-light interaction regime as the magnetisation recovers to a polarised ferromagnetic (FM) state after a few picoseconds. As the fluence increases within the range of 0.055–0.087 mJ cm$^{-2}$, the pulse significantly demagnetises the system, creating densely packed magnon droplets transiting the system to the formation of non-trivial objects such as spin stripes (Supplementary Movie S3). The temperature equilibration in this case is not high enough to break the stripes in more localized spin quasiparticles. The formation of skyrmions however can be achieved as the energy is slightly increased to 0.09–0.180 mJ cm$^{-2}$. At this limit, the temperature of the system is suitable for the magnon droplets to gain topological features, and hence they evolve into skyrmions rather than merging (Fig. 2b, Supplementary Movie S4). It is noteworthy that is within the region of 0.14–0.18 mJ cm$^{-2}$ there is the simultaneous formation of skyrmioniums with skyrmions which are randomly distributed over the surface. At fluences above 0.18 mJ cm$^{-2}$ (Fig. 2c, Supplementary Movies S5–S6), a hybrid spin state consisting of skyrmions, stripes and skyrmioniums are formed. The large amount of energy deposited in the CrGeTe$_3$ layers at this limit results in thermal fluctuations which are sufficient to inhibit the creation of magnon droplets. Instead, the non-equilibrium spin textures tend to merge and evolve in mixed labyrinthine domains with the appearance of skyrmions and skyrmioniums. In all these different processes, the magnitude of $N_{sk}$ can work as a descriptor indicating the different spin objects formed by the excitations. However, visualisation of the spin distributions is critical to further characterise their spatial correlation and interactions.

We can experimentally validate the formation of spin textures and further tuning with the laser pulses via WFKM techniques (see Methods for details). We initially zero-field cooled the system to ≈ 18.5 K, and then applied a sequence of $N = 1$, 10, and 100 pulses under an out-of-plane field of 25 mT. Figure 2d, e show the spin states obtained in CrGeTe$_3$ after $N = 100$ and $N = 10$ pulses, respectively. Strikingly, these results follow closely those obtained in the simulations (Fig. 2b, c) with the formation of magnetic bubbles and their coexistence with magnetic stripes throughout the surface. The nominal fluence used in the measurements (16 mJ/cm$^2$) agrees on the hybrid spin structures found for magnitudes above 0.30 mJ/cm$^2$ in the phase diagram in Fig. 2a. This indicates a complex scenario for the stabilisation of the spin textures with the laser excitations. We also noticed that a high number of pulses is normally necessary in the measurements to induce the creation of the spin textures (Fig. 2d, e). That is, for $N \leq 10$ we barely observed any modifications of the spin textures at the same laser fluence. As discussed previously, this is related with the number of magnon droplets populating the surface during the magnon localisation phenomenon[60]. A closer look at the magnetisation dynamics reveals that depending on the amount of energy transferred to the layers via the different number

optical excitations. Domain wall widths were extracted and resulted in ~ 5.2 nm which follows those found in strong 2D ferromagnets such as CrI$_3$[4], CrBr$_3$[6], and Fe$_3$GeTe$_2$[45,57]. It is worth mentioning that the formation of skyrmioniums into the system is more efficient via ultrafast laser pulses than via cooling. We have tested our simulation setup and found that statistically one out of four repeats of a field-cooling LLG-simulations (11.7 mT) yielded the formation of skyrmioniums. This suggests that excited ground states are be more feasible for the stabilisation of spin textures since any energy barrier for the formation might be overcome[54].

The formation of spin textures in CrGeTe$_3$ after the laser pulses can be explained in terms of the spin correlations present in the system that do not vanish during the initial rapid quenching of the magnetisation (<250 fs) during the demagnetisation process[58,59]. Following the

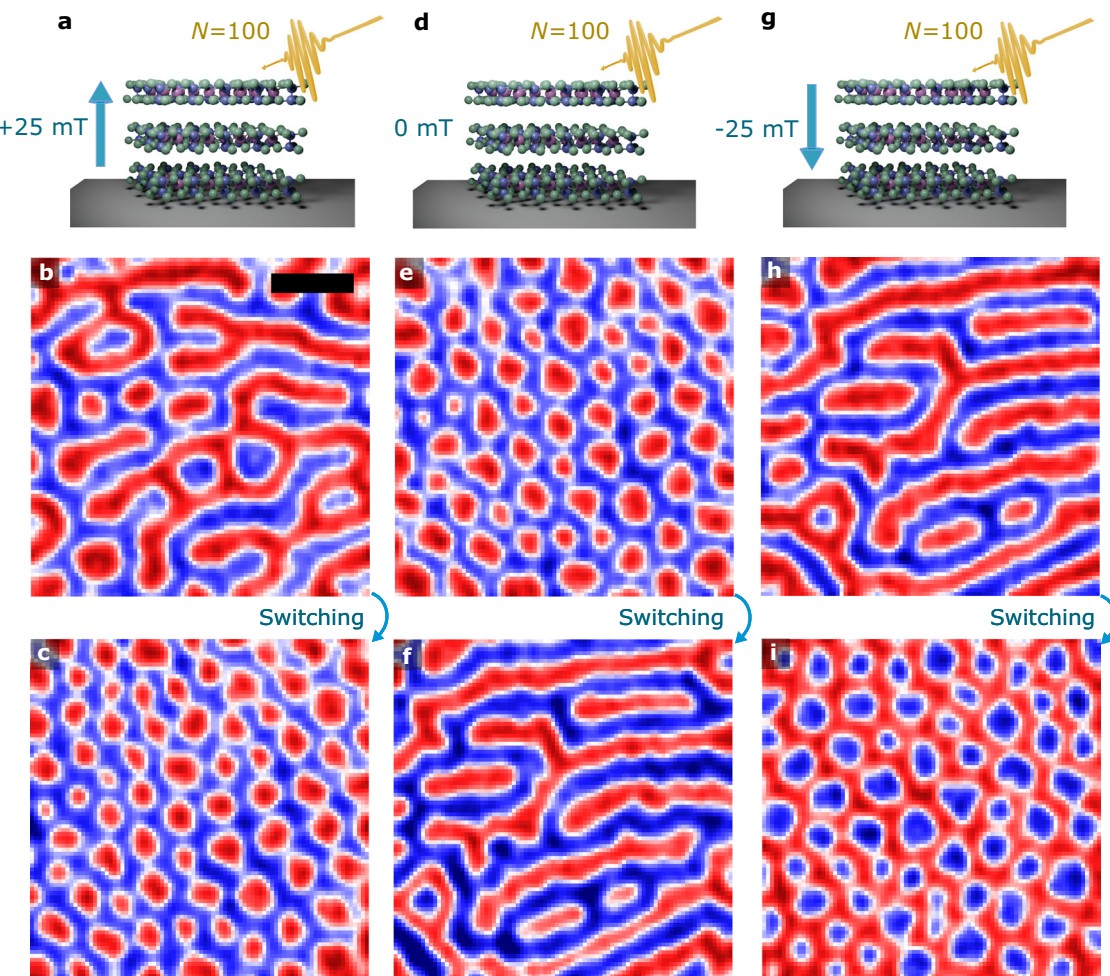

**Fig. 3 | Light-induced spin toggle switch. a** Schematic of CrGeTe$_3$ crystal under laser excitations ($N = 100$ pulses) and an out-of-plane magnetic field (+25 mT). **b–c,** Spin toggle switching observed between two magnetic configurations dominated by stripe domains (**b**) and bubbles/skyrmions (**c**) via WFKM measurements. A magnetic field of + 25 mT and $N = 100$ laser pulses are applied to induce the transition. **d** Similar as in **a** but without any applied field. **e–f** The magnetic configuration in **c** is used as a reference state to a new switching process in **e** with zero field and under $N = 100$ laser pulses. The final equilibrated state is shown in **f** with a predominant amount of stripe domains throughout the surface. **g** Similar as in **a** but with a magnetic field of opposite polarity (−25 mT). **h,i** The magnetic state in **f** is used as the reference state in **h** to induce a new switching step under −25 mT and $N = 100$ laser pulses. The reversal of the field from + 25 mT (**a–c**) to −25 mT (**g–i**) with an intermediate step at 0 mT can generate a full switching loop with a different polarisation of the magnetic skyrmions and stripe domains. The scale bar is 5 μm and the laser fluence is 16 mJ/cm². The measurements were performed at 18.5 K.

of pulses $N$, the magnon droplets can either be isolated and give place to the formation of skyrmions or magnetic bubbles; or, they can coalesce and form long stripe domains with smaller contributions from localised droplets. We noticed that both situations are possible in CrGeTe$_3$ depending on the number of pulses applied under an external magnetic field. Moreover, the size of the magnetic spin textures stabilised in the areas studied (~1–2.90 μm) exceeded those calculated in the atomistic computations (~17–29 nm). However, we have observed smaller spin textures (Supplementary Fig. 4) within the range of the simulations (~20–60 nm) which could not be resolved due to optical limitations of the technique. Higher resolution techniques, e.g., Lorentz transmission electron microscopy, would be required to resolve such small-diameter textures.

An outstanding question raised by these results is whether the ultrafast laser pulses can induce the transformation of magnetic stripes into bubbles/skyrmions, and vice-versa, like in a spin-toggle switch[13]. This AOS process normally requires two magnetic states with equal energies where the switching between them can be realised at a negligible heat generation[61,62]. The switching however needs to be reproducible and reversible in order to allow full writing of the magnetic information[62,63]. To investigate this we probed the variation

of the spin textures with laser excitations but also taken into account an external field to unveil any polarisation effect that might be present in the system. As laser pulses are applied to CrGeTe$_3$, there are clear transformations of the magnetic textures from one state to another which can be further manipulated via a magnetic field (Fig. 3). We considered three steps in this switching phenomenon under a field of +25 mT (Fig. 3a–c), 0 mT (Fig. 3d–f) and −25 mT (Fig. 3g–i). We initially obtained a spin state with mixed labyrinthine domains (Fig. 3b) prior laser application, which will be used as a reference for the switching process. We then applied $N = 100$ laser pulses at +25 mT on this state to generate a switching between a stripe dominant-state (Fig. 3b) to a random network of magnetic bubbles (Fig. 3c). We continued this process through magnetic fields of 0 mT (Fig. 3d–f) and −25 mT (Fig. 3g–i) to show that this toggle switching is reversible, complete and reproducible. It is noticed that the polarity reversal of the spin textures with magnetic fields (Fig. 3c, i) provides an additional ingredient for controlling on-demand the polarisation of the spin textures or to determine which spin textures are created. For instance, the switch from magnetic bubbles to stripe domains occurs without the application of magnetic fields. However, the transition from stripes to bubbles has been noticed to require a

bias field. This indicates a switch mechanism where a defined spin state can be obtained through the interplay between laser pulses and field-induced symmetry breaking into the lattice. Simulation results of the toggle switch found in the experiments are given in Supplementary Fig. 2 which provided a theoretical background for our observations.

## Discussion

The discovery of laser-induced spin switching in 2D vdW magnetic materials creates exciting avenues for imprinting and tailoring topological properties at the atomic level in layered systems. Several compounds such $CrGeTe_3$ have been explored, isolated and integrated in devices geometries[9] which could be applied to the exploration of ultrafast spintronics. Our results suggest that even center-symmetric lattices but with sizeable non-collinear asymmetric exchange (e.g., DMI) in the second nearest-neighbors[47] may be able to hold similar features. The ultrafast laser-induced heating of electrons in $CrGeTe_3$ is able to trigger magnetization control which liaised with external magnetic fields create a spin switch mechanism. Different field polarisation also controls the polarity of the created spin textures (e.g., stripes, bubbles, skyrmions, anti-skyrmions) as we compare the final spin state with fields of different signs (Fig. 3). Indeed, the skyrmions are stable after removal of the magnetic field and can be switched back to the initial stripe pattern by exposing the material to the same number of pulses without applying any external field. This gives further flexibility to tune the spin configurations on-demand without constraint on the setup to be coupled to large magnets. The number of pulses $N$ however still needs to be large relative to single-pulse toggle switch in 3D ferrimagnets[62,64] in order to observe phase transformation of the magnetic textures (e.g., skyrmions into stripe domains, and/or vice-versa) or induce the initial formation of topologically non-trivial spin objects. Even though such difference could be seen as a limitation in terms of single-pulses applications, it suggests that if a 2D ferrimagnet could be isolated and chemically stabilised such system could potentially be probed using the guidelines demonstrated in our work. This opens a broad range of material possibilities to be explored where fast high-throughput screening can be used to select potential candidates. Moreover, to be able to write information and switch magnetization between different spin states as those demonstrated on $CrGeTe_3$, the laser energy plays an important role. The challenge therefore is to find a balance where minimum energy fluence can be used whereas allowing efficient interaction between spins, light and crystal symmetry towards topological modifications. It is worth mentioning that the reversible topological switch mechanism reported here is different to that shown in Pt/CoFeB/MgO and Pt/Co systems[65]. On that, just one-way switching (either from a uniform magnetisation state, or stripe domains, into skyrmions, but not vice-versa) was found. In this aspect, the finding of a fully reversible topological spin switch based on a 2D vdW magnet without any interface issues or elaborate sample preparation presents as a leap onward writing/reading topological information. The natural next-step however is how to encode these magnetic objects with specific information (e.g., bytes) and read them afterwards. Much work is needed to fully explore the whole sets of switch phenomena at strictly 2D compounds.

## Methods

### Atomistic simulations

We model the system through atomistic spin dynamic simulations methods[4,42,56]. Spin interactions are described via Eq. (1) with parameters from ab initio calculations[2]. The Landau-Lifshitz-Gilbert (LLG) equation is used to describe the dynamics at different times and temperatures. The LLG equation is a differential equation that describes the precessional motion of magnetisation in a crystal. It is

given by:

$$\frac{\partial \mathbf{S}_i}{\partial t} = -\frac{\gamma}{(1+\lambda^2)}\left[\mathbf{S}_i \times \mathbf{B}_{\text{eff}}^i + \lambda \mathbf{S}_i \times (\mathbf{S}_i \times \mathbf{B}_{\text{eff}}^i)\right] \quad (3)$$

where $\mathbf{S}_i$ is a unit vector describing the spin moment orientation of site $i$. $\gamma$ is the ratio of a spin's magnetic moment to its angular momentum, known as the gyromagnetic ratio, and $\mathbf{B}_{\text{eff}}^i$ is the effective net magnetic field. This effective field can be derived from the first derivative of the spin Hamiltonian:

$$\mathbf{B}_{\text{eff}}^i = -\frac{1}{\mu_S}\frac{\partial \mathscr{H}}{\partial \mathbf{S}_i} \quad (4)$$

where $\mu_S$ is the local spin moment. The Heun method is used to numerically integrate the LLG equation[42].

### Two-temperature model

The semiclassical two-temperature model (2TM)[43,49] was utilised to simulate the thermal transport during the ultrafast laser heating on $CrGeTe_3$. We previously modeled the laser-induced spin texture formation in $CrCl_3$[43] using this technique. We did not observe any spin-toggle switch as that found in $CrGeTe_3$, but rather the formation of merons or anti-merons from a homegeneous magnetic state. This suggests that the intrinsic magnetic properties of the vdW layer (e.g., easy-plane or easy-axis) play a role in the switching phenomena. The 2TM separates the temperature of the system into electron and phonon (lattice) contributions represented by $T_{\text{elec}}$ and $T_{\text{phon}}$, respectively. The model assumes that internal relaxations are faster then the coupling heat baths which allow to describe the interactions between lattice and electrons via coupled differential equations given by:

$$m\frac{\partial \overline{v}}{\partial t} + m\overline{v}\cdot\nabla_r\overline{v} + \left[k_b\left(1 + \frac{T_{\text{elec}}}{C_e}\frac{\partial C_e}{\partial T_{\text{elec}}}\right) - e\beta\right]\nabla T_{\text{elec}} = -\frac{eT_{\text{elec}}\overline{v}}{\mu_0 T_{\text{phon}}} \quad (5)$$

$$C_e\left(\frac{\partial T_{\text{elec}}}{\partial t} + \overline{v}\cdot\nabla_r T_{\text{elec}} + \frac{2}{3}T_{\text{elec}}\nabla_r\cdot\overline{v}\right) + \nabla_r\cdot\overline{Q}_e = -G(T_{\text{elec}} - T_{\text{phon}}) + S(\overline{r},t) \quad (6)$$

where $m$ is the electron mass, $\overline{v}$ is the mean (drift) velocity vector, $k_b$ is the Boltzmann constant, $C_e$ is the electron heat capacity constant, $\beta$ is the electric field coefficient, $e$ is the electron charge, $\mu_0$ is the mobility of electrons at room temperature, $\overline{Q}_e$ is the electronic heat flux, $G$ is the electron-phonon coupling, and $S(\overline{r},t)$ is the volumetric laser heat source. Equations (5)–(6) describe the conservation of momentum and energy in the electron subsystem. The description of the lattice is given by:

$$C_l\frac{\partial T_l}{\partial t} = -\nabla\cdot\overline{Q}_l + G(T_e - T_l) \quad (7)$$

where $C_l$ is the lattice heat capacity, and $\overline{Q}_l$ is the lattice heat flux. The coupling between lattice and electronic subsystems is given via:

$$\tau_e\frac{\partial \overline{Q}_e}{\partial t} + \overline{Q}_e = -K_e\nabla T_e \quad (8)$$

$$\tau_l\frac{\partial \overline{Q}_l}{\partial t} + \overline{Q}_l = -K_l\nabla T_l \quad (9)$$

where $\tau_e$ is the electron relaxation time, $K_e$ is the electronic thermal conductivity coefficient, $\tau_l$ is the lattice relaxation time, and $K_l$ is the lattice thermal conductivity coefficient. The parametrization of the 2TM is fitted accordingly to experimental magnetisation dynamics on

CrGeTe$_3$ and parent compounds[66–68] as shown in Supplementary Table 3. The use of 2TM liaised with the LLG equation allows the simulations of systems for several nanoseconds (>8 ns) at large areas (e.g., 250 × 250 nm$^2$) which are unpractical using other techniques, for instance, time-dependent density functional theory[69] (TDDFT). Normally TDDFT is well suited for simulating attosecond to picosecond phenomena which occurred immediately after the photo-excitation of the system (i.e., charge transfer pathways)[70]. The time-step in such simulations is generally within the sub-attosecond timescale (1 attosecond = $10^{-18}$ s) as high-frequency fluctuations arise from the electron dynamics included in the time-dependent Kohn-Sham equation[69]. In our case however the critical equilibration where the topological spin textures are observed takes place long after the laser pulses (>1 ns), and no electronic effects are present apart from the thermal electronic bath provided via the 2TM. The sound agreement between the experimental results and the simulations indicates the LLG-2TM as a suitable approach to study photon-excitation in 2D magnets.

## Topological number

Calculations of the topological charge have been used to identify different spin textures in the CrGeTe$_3$. By convention skyrmions have a topological charge $N_{sk} = -1$ while antiskyrmions have $N_{sk} = +1$. Trivial bubbles have a trivial topological charge $N_{sk} = 0$, similarly as skyrmioniums. Note that the topological charge depends on both chirality and polarity; reversal of either will correspondingly reverse the sign of $N_{sk}$. In this convention a core-down, right-handed skyrmion has a topological charge of $N_{sk} = -1$. In the continuum case the topological charge is defined as in Eq. (2). In the discrete approach, we calculate $N_{sk}$ via the triangulation method[37] where the spin texture lattice is partitioned into triangles involving the spins and the sum is evaluated over the whole surface. The calculation is performed 3 ns of relaxation after the laser pulse.

## Samples

Single-crystalline CrGeTe$_3$ (CGT) flakes were synthesized using the chemical vapor transport method. Elemental precursors of chromium (≥99.995%), germanium (≥99.999%), and tellurium (99.999%) in the molar ratio of Cr:Ge:Te = 10:13.5:76.5 were sealed in a thick-walled quartz ampule evacuated by the turbomolecular pump down to ~10-5 mbar. Excess tellurium was added as a flux. To ensure the high purity of the product, no other transport agents were used. Once sealed, the ampule was thoroughly shaken to mix the precursors. Then, the ampule was loaded into a horizontal two-zone annealing furnace with both zones slowly ramping to 950 °C. The ampule was kept at 950 °C for 1 week and then slowly cooled (0.4 °C/h) with a temperature gradient between two zones of the furnace. The gradient ensured the crystallization predominantly at the cold end of the ampule. Once the furnace reached 500 °C, the heaters at both zones were switched off allowing the furnace to cool naturally to room temperature. The crystals were extracted from the ampule under inert conditions and stored for future use. Our samples are ~1 mm thick, and 1.5 × 1.5 mm for lateral dimensions. Supplementary Figs. S3–S5 show the results for the characterisation on x-ray diffraction patterns, domain structures obtained via WFKM and hysteresis loop, respectively. Supplementary Movies S7 and S8 provide additional details on the measurements.

## Microscopy

The measurements were performed by wide-field Kerr microscopy (WFKM) in a polar geometry to sense the out-of-plane magnetization in response to either a magnetic field or optical pulses. The sample illumination was linearly polarized, while polarization changes of the reflected light due to the polar Kerr effect were detected as intensity changes using a nearly crossed analyzer, quarter-waveplate, and high sensitivity CMOS camera. For switching experiments, an optical pump beam (1035 nm) with linear polarization and 300 fs pulse duration, 1 MHz repetition rate and different number of pulses $N$ was incident at 45° to the sample plane and focused to a 100 $\mu$m diameter spot (intensity at $1/e^2$). Measurements were performed at temperatures ranging from 15 to 50 K. All images presented in this work are acquired for the final magnetization state which remains the same until the sample is exposed either to more optical pulses, or to an external magnetic field.

## Data availability

The data that support the findings of this study are available within the paper, Supplementary Information and upon reasonable request.

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

## Acknowledgements

M.D., P.S.K., and R.J.H. gratefully acknowledge the support of the Engineering and Physical Sciences Research Council (EPSRC) through grants EP/V054112/1 and EP/V048538/1. H. K. and S.K. acknowledge support from EPSRC on EP/T006749/1. G.E. acknowledges support from the Ministry of Education (MOE), Singapore, under AcRF Tier 3 (MOE2018-T3-1-005) and the Singapore National Research Foundation for funding the research under medium-sized centre program. EJGS acknowledges computational resources through CIRRUS Tier-2 HPC Service (ec131 Cirrus Project) at EPCC funded by the University of Edinburgh and EPSRC (EP/P020267/1); ARCHER UK National Supercomputing Service (http://www.archer.ac.uk) via Project d429, and the UKCP consortium (Project e89) funded by EPSRC grant ref EP/P022561/1. EJGS acknowledge the Spanish Ministry of Science's grant program "Europa-Excelencia" under grant number EUR2020-112238, and the EPSRC Open Fellowship (EP/T021578/1) for funding support. For the purpose of open access, the author has applied a Creative Commons Attribution (CC BY) licence to any Author Accepted Manuscript version arising from this submission.

## Author contributions

E.J.G.S., H.K., and R.H. conceived the project. M.K. performed the simulations under the supervision of E.J.G.S. M.D., S.K., and P.S.K. performed the experiments. M.D., S.K., R.H., H.K., and E.J.G.S. analysed the experimental data. I.V. and G.E. provided the samples and characterisation. E.J.G.S. wrote the paper with inputs from all authors. All authors contributed to this work, read the manuscript, discussed the results, and agreed on the included contents.

## Competing interests

The authors declare no competing interests.
