## [Peer Review File · Nature Communications]

Reviewers' Comments:

Reviewer #1:

Remarks to the Author:

The manuscript entitled "Laser-induced topological spin switching in a 2D van der Waals magnet" by Maya Khela et al describe their micromagnetic simulation and Kerr images of laser-induced topological spin switching in CrGeTe₃. The interplay of light and spin in vdW magnets is an interesting issue. The manuscript reported an interesting study not only on ultrafast generation of skyrmions, anti-skyrmions, skyrmioniums, and stripe domains, but also a controllable and reversible transformation between these spin textures. Overall, the conclusions are quite remarkable. However, there are several comments I must point out.

1, It is well known for that CrGeTe₃ is centrosymmetric and DMI disappears. I notice that the authors cited this theoretical paper (PRB 102, 241107 (2020)). In this paper, DMI appears in CrGeTe₃ only when it is monolayer. More, this predication still needs be confirmed by experiments. For the sample used in the manuscript, I can't believe the existing of DMI without more supporting results.

2, the characteristic sizes of spin texture obtained from the simulation (~ 10 nm) is nearly two orders of magnitude smaller than that from Kerr image (~1 micron). The skyrmion sizes in many materials have been predicated quite well with the experimental results. In here, this big gap is hard to understand if holding DMI. On the anther hand, there are many reports that the bubble and stripe domain still can be generated without DMI. From this point, is DMI necessary?

From above two points, I think the simulation results are suitable for the materials with DMI, not for CrGeTe₃. I can't recommend the publication of this manuscript in Nature Communications.

Reviewer #2:

Remarks to the Author:

The paper reports about various spin textures that can be induced in a 2D vdW material under influence of femtosecond laser pulses. In particular the paper describes the pattern of magnetic domains in CrGeTe₃ gets modified after excitation with a sequence of sub-ps laser pulses with and without magnetic field. The goals of the work are directly related to such hot topics in contemporary condensed matter physics as ultrafast magnetism, skyrmions, vdW materials. Hence I have no doubts that the paper will be noticed by specialists in several rapidly developing, booming fields of magnetism. The work combines simulations and experimental results obtained with the help of magneto-optical microscope. The paper is written exceptionally well. I have only very few important comments, which the authors should be able to address.

- Page 6. I notice strange temperature in K given between parathesis - 0.34 K and 2.25 K. Are these temperatures show temperature increase after an individual laser pulse? If yes, I do not understand why such a small values of temperature increase can lead to any substantial changes of magnetization (demagnetization drive by phonon temperature). Note that the measurements are done at 18 K and the Curie temperature is 68 K. -
- Nowhere in the manuscript I could find info about thickness of the studied flakes.
- even very trivial, but still very important characterization of the studied samples is missing. Here I mean temperature dependence of the magneto-optical effect and hysteresis loops. This info would be very useful for those specialists who will be willing to reproduce the results.

Reviewer #3:

Remarks to the Author:

This paper explores the optical control of topological spin switching in 2D vdW magnets. By combing wide-field Kerr microscopy experiments and a multiscale approach (DFT + spin model), the authors show an efficient switching of topological spin structures (skyrmion, anti-skyrmion, skyrmionium) under the laser and external magnetic fields.

The results are scientifically sound, but I believe that it lacks enough novelty for publication in

Nature Communications. In particular, similar results have been extensively discussed by some of the authors in recent publications:

1. M. Dąbrowski et al., "All-optical control of spin in a 2D van der Waals magnet", Nat. Commun. 13, 5976 (2022), the same "theory + experiment"
2. M. Strungaru et al., "Ultrafast laser-driven topological spin textures on a 2D magnet", npj Computational Materials, 169 (2022), the same theory

Therefore, I recommend the authors seek publication in a more specialized journal.

Several comments are listed below:

- I would suggest changing "higher-order exchange interactions" to "biquadratic exchange interaction". As far as I know, biquadratic exchange interaction is a particular situation of 4-site 4-spin interaction. It turns out that 4-site 4-spin interaction is crucial for the skyrmion stability in ultrathin films as discussed in S. Paul et al., Nat. Commun. 11, 4756 (2020).
- It is not clear to me how different magnetic interactions change according to the laser. I would recommend the authors present explicitly these details, which are essential for spin dynamics simulations.

REVIEWER COMMENTS

Reviewer #1 (Remarks to the Author):

The manuscript entitled “Laser-induced topological spin switching in a 2D van der Waals magnet” by Maya Khela et al describe their micromagnetic simulation and Kerr images of laser-induced topological spin switching in CrGeTe₃. The interplay of light and spin in vdW magnets is an interesting issue. The manuscript reported an interesting study not only on ultrafast generation of skyrmions, anti-skyrmions, skyrmioniums, and stripe domains, but also a controllable and reversible transformation between these spin textures. Overall, the conclusions are quite remarkable. However, there are several comments I must point out.

Response 1: *We thank the Reviewer for the kind comments about our manuscript, and for considering our work interesting, with quite remarkable conclusions. We have addressed below all points raised by the Reviewer and modify the manuscript accordingly.*

1, It is well known for that CrGeTe₃ is centrosymmetric and DMI disappears. I notice that the authors cited this theoretical paper (PRB 102, 241107 (2020)). In this paper, DMI appears in CrGeTe₃ only when it is monolayer. More, this predication still needs be confirmed by experiments. For the sample used in the manuscript, I can't believe the existing of DMI without more supporting results.

Response 2: *We thank the Reviewer for the comments. At the level of first-nearest neighbours (1st-NNs) CrGeTe₃ is a centrosymmetric material where DMI is strictly zero. However, at the level of second-nearest-neighbours (2nd-NNs) there is a break of the spatial inversion symmetry following the Moriya's rules, which generated a sizeable DMI into the system. In particular, this is responsible for the opening of a band gap at the K-point in CrGeTe₃ recently measured using inelastic neutron scattering (Zhu et al., Sci. Adv. 2021; 7: eabi7532). The value of DMI estimated from these measurements is 0.32 meV which corresponds to the value calculated ~0.31 meV (PRB 102, 241107 (2020)) and used in our simulations. Indeed, we had included in the manuscript consideration of DMI at 2nd-NNs at the Discussion section (page 12, lines 222-224).*

We have included additional comments highlighting the inclusion of DMI at 2nd-NNs in our simulations and additional references (page 5, lines 87-90) supporting our arguments.

2, the characteristic sizes of spin texture obtained from the simulation (~ 10 nm) is nearly two orders of magnitude smaller than that from Kerr image (~1 micron). The skyrmion sizes in many materials have been predicated quite well with the experimental results. In here, this big gap is hard to understand if holding DMI. On the anther hand, there are many reports that the bubble and stripe domain still can be generated without DMI. From this point, is DMI necessary?

Response 3: *We thank the Reviewer for the comments and question. The difference in size between calculated and observed skyrmions is due to: 1) the limitations of the atomistic spin dynamics methods, which can provide a robust picture of the formation of the different spin textures in CGT, but at a smaller scale relative to the experiments; 2) the material parameters used in our simulations which can incidentally change the size of the skyrmions as discussed in Ref.49. Indeed, we mentioned in the manuscript that we assumed a more qualitative approximation in the comparison of the diameters found in simulations with those observed in the measurements; and, more importantly, 3) in the measurements we have also observed*

smaller spin structures (<20-60 nm) than those showed in Figure 2-3 which could not be resolved due to optical limitations of the technique. **Figure R3** shows an example of CGT at different temperatures (60-12 K) where small-diameter spin textures can clearly be observed between the large dark structures but cannot be optically resolved. The range of those spin textures agrees well with that used in our atomistic spin dynamic simulations. We appreciated the Reviewer's comment on the comparison between simulations and experiments. However, to resolve small spin textures would require higher resolution methods, e.g., Lorentz TEM, using liquid Helium (due to the low T_c of CGT) which are beyond the scope of the present study.

Figure R1: Snapshots of the zero-field cooling spin dynamics including all the interactions considered in Eq. 1 in the main text of the manuscript except DMI, which is set to zero. The magnetization is projected along the out-of-plane component (M_z), and an in-plane one (M_x). Colour code shows the different polarisation of the spins.

We have emphasized these arguments in the updated version of the text (pages 7-8, lines 122-127; page 11, lines 190-193).

Regarding the formation of bubbles and stripe domains, we agree with the Reviewer that DMI is not needed. **Figure R1** shows snapshots of the spin dynamics at different times during zero-field cooling including exchange interactions, dipolar interactions, biquadratic exchange but excluding DMI. Several stripe domains are formed along the out-of-plane component of the magnetisation (M_z) whereas topologically trivial magnetic bubbles are formed along the in-plane components (e.g., M_x).

Moreover, as it is mentioned in the manuscript (page 5, lines 83-86), we have undertaken a careful study of the role of different spin interactions (e.g., DMI, biquadratic exchange, dipolar interactions) in the stabilisation of skyrmions in CrGeTe_3 . A summary of all these time-consuming simulations were included at Supplementary Table S1 with the key conclusion that DMI is a necessary ingredient for the stabilisation of skyrmions into the system. Without it, none of the topological spin textures observed in the simulations are energetically stable against thermal fluctuations.

We have included a new simulation movie in SI where **Figure R1** was extracted showing

the zero-field cooling spin dynamics including all the interactions considered in Eq. 1 except DMI, which is set to zero.

From above two points, I think the simulation results are suitable for the materials with DMI, not for CrGeTe₃. I can't recommend the publication of this manuscript in Nature Communications.

Response 4: *We thank very much the Reviewer for evaluating our manuscript. We have considered each point raised by the Reviewer and modify the manuscript accordingly showing strong scientific arguments that support our conclusions. We univocally demonstrated the presence of DMI in CrGeTe₃ via published inelastic neutron scattering experimental results, symmetry arguments and simulation data. We hope the Reviewer will reconsider his/her position.*

Reviewer #2 (Remarks to the Author):

The paper reports about various spin textures that can be induced in a 2D vdW material under influence of femtosecond laser pulses. In particular the paper describes the pattern of magnetic domains in CrGeTe₃ gets modified after excitation with a sequence of sub-ps laser pulses with and without magnetic field. The goals of the work are directly related to such hot topics in contemporary condensed matter physics as ultrafast magnetism, skyrmions, vdW materials. Hence I have no doubts that the paper will be noticed by specialists in several rapidly developing, booming fields of magnetism. The work combines simulations and experimental results obtained with the help of magneto-optical microscope. The paper is written exceptionally well. I have only very few important comments, which the authors should be able to address.

Response 5: *We thank the Reviewer for evaluating our manuscript and for the kind words regarding our work. We have considered each point raised by the Reviewer and modified the manuscript accordingly.*

- Page 6. I notice strange temperature in K given between parathesis - 0.34 K and 2.25 K. Are these temperatures show temperature increase after an individual laser pulse? If yes, I do not understand why such a small values of temperature increase can lead to any substantial changes of magnetization (demagnetization drive by phonon temperature). Note that the measurements are done at 18 K and the Curie temperature is 68 K.

Response 6: *We thank the Reviewer for the comments and question. The temperatures forementioned correspond to the equilibration temperatures obtained after the system is excited by the laser pulse and thermally equilibrated. That is, we initially observed a sharp increment of the temperature (electronic) beyond the Curie temperature after the laser pulse hit the system, which is followed by the phononic temperature after a few femtoseconds. Both temperatures (electronic, phononic) time-evolved thermodynamically until equilibration is stabilised.*

We have included all calculated values at different fluences in Supplementary Table S3 in SI.

- Nowhere in the manuscript I could find info about thickness of the studied flakes.

Response 7: *We thank the Reviewer for pointing this out. Our samples are ~1 mm thick, and 1.5 x 1.5 mm for lateral dimensions. We have included these details in the manuscript (page 16, lines 272-273).*

- even very trivial, but still very important characterization of the studied samples is missing. Here I mean temperature dependence of the magneto-optical effect and hysteresis loops. This info would be very useful for those specialists who will be willing to reproduce the results.

Response 8: We thank the Reviewer for the suggestion. Our samples follow the same characterisation as in Khan et al. *Phys Rev. B* 100, 134437 (2019). We have included the following additional results in SI to fully address the Reviewer's comments: i) X-ray diffraction results for the used CGT crystals (**Figure R2**), ii) temperature dependent domain structure images acquired via with the wide-field Kerr microscopy (as described in the Methods) (**Figure R3**), iii) hysteresis loop at 6 K extracted from the domain structure images acquired with beam-scanning Kerr microscopy (**Figure R4**), and iv) Supplementary Movies S7 and S8 show the magnetisation reversal using WFKM and beam-scanning microscopy, respectively. It is worth mentioning that although the WFKM provides better quality domain structure images, the maximum available field (100 mT) is not sufficient to fully reverse the magnetisation. This is one of the reasons we have performed additional measurements with the beam-scanning microscopy to support the WFKM dataset. This new dataset is mentioned in the main text at page 16, lines 273-275.

Figure R2: X-ray diffraction patterns of pristine bulk crystal of CrGeTe₃, used in the measurements. The plot shows a series of out of plane (00l) peaks which agrees well with the earlier report in the literature (e.g., Khan et al. *Phys Rev. B* 100, 134437 (2019)).

Figure R3: *a-f*, Domain structure images at different temperatures observed by wide-field Kerr microscopy (WFKM) (as described in Methods) in a polar geometry sensitive to the out-of-plane magnetization. The sample illumination was linearly polarized LED white light, while polarization changes of the reflected light due to the polar Kerr effect were detected as intensity changes using a nearly crossed analyser and quarter-waveplate. The measurements were performed at remanence and within the temperature range 65-12 K. *g*, Variation on the polarisation for spin up and down as a function of the temperature at the area highlighted in *f*.

Figure R4: The hysteresis loop extracted from domain structure images acquired with beam-scanning Kerr microscopy using microstat MO 5T superconducting magnet, with the field direction applied perpendicular to the sample surface. The sample illumination was linearly

polarized CW green laser diode, while polarization changes of the reflected light due to the polar Kerr effect were detected using a balanced polarizing photodiode bridge detector. The measurements were performed at 6 K. The domain structure evolution during the magnetization reversal from $-H$ (-250m T) to $+H$ (+250mT) (red circles) can be observed in the Supplementary Movie S8.

Reviewer #3 (Remarks to the Author):

This paper explores the optical control of topological spin switching in 2D vdW magnets. By combining wide-field Kerr microscopy experiments and a multiscale approach (DFT + spin model), the authors show an efficient switching of topological spin structures (skyrmion, anti-skyrmion, skyrmionium) under the laser and external magnetic fields.

The results are scientifically sound, but I believe that it lacks enough novelty for publication in Nature Communications. In particular, similar results have been extensively discussed by some of the authors in recent publications:

1. M. Dąbrowski et al., "All-optical control of spin in a 2D van der Waals magnet", Nat. Commun. 13, 5976 (2022), the same "theory + experiment"
2. M. Strungaru et al., "Ultrafast laser-driven topological spin textures on a 2D magnet", npj Computational Materials, 169 (2022), the same theory

Therefore, I recommend the authors seek publication in a more specialized journal.

Response 9: *We thank the Reviewer for his/her comments, and to mention that our results are scientifically sound. We appreciated that the Reviewer mentioned these two recent manuscripts from our groups which set the ground for several landmark achievements. However, in neither of them it is demonstrated that the spin textures can be switched between two magnetic ordered states (e.g., stripe domains into skyrmions, and vice-versa), nor the reversible control of their spin polarisation via the interplay of laser excitations and magnetic fields. Our results in this context are timely, novel and open a new pathway for the control of topological spin objects in vdW magnets not reported in previous publications. It is worth mentioning that it is the first time, to the best of our knowledge, a topological spin switch is demonstrated in a 2D magnetic material which clearly broaden the scope for switching processes and topological manipulation. Furthermore, the two mentioned manuscripts concern two different 2D magnetic materials (CrI_3 , CrCl_3) which are remarkably distinct in magnetic properties relative to CGT.*

Moreover, as pointed out by #Reviewer 2: "I have no doubts that the paper will be noticed by specialists in several rapidly developing, booming fields of magnetism." Hence, we are confident that our findings will be well received by the community working on these hot subjects.

Several comments are listed below:

- I would suggest changing "higher-order exchange interactions" to "biquadratic exchange interaction". As far as I know, biquadratic exchange interaction is a particular situation of 4-site 4-spin interaction. It turns out that 4-site 4-spin interaction is crucial for the skyrmion stability in ultrathin films as discussed in S. Paul et al., Nat. Commun. 11, 4756 (2020).

Response 10: *We thank the Reviewer for the comments and suggestions. We have modified the text to highlight the inclusion of biquadratic exchange interactions instead of higher order*

exchange interactions. We have also included a citation to the manuscript mentioned by the Reviewer together with a discussion on the importance of 4-site 4-spin interaction on the stability of skyrmions on ultrathin films (page 5, lines 79-83).

- It is not clear to me how different magnetic interactions change according to the laser. I would recommend the authors present explicitly these details, which are essential for spin dynamics simulations.

Response 11: *We thank the Reviewer for pointing this out. In the simulation of the laser-induced spin dynamics on CrGeTe₃ via the two-temperature model (Ref. 45), the spin interactions (e.g., bilinear exchange, anisotropic exchange, biquadratic exchange, DMI, single-ion anisotropy) are not modified by the excitations, but rather the laser changes the spin dynamics under different fluences (Figure 1). This is an effect of the time-evolution of the system to minimize the energy towards thermal equilibration. Nevertheless, these spin interactions (e.g., magnetic parameters) remain fixed throughout the dynamics. The sound agreement with the experimental results indicates that this hypothesis is correct and fully support our analysis. Moreover, variations of the different magnetic interactions with the laser excitations are currently unknown in CrGeTe₃ and clearly beyond the scope of our study.*

We have included additional discussion of the parameters used in the simulations, and clarified their role in the excited-state spin dynamics (page 6, lines 93-97).

Reviewers' Comments:

Reviewer #1:

Remarks to the Author:

The authors have addressed all my comments. And I would like to recommend the publication in NC.

Reviewer #2:

Remarks to the Author:

The authors have adequately addressed my concerns in the revised version of the manuscript. Although the other two Referees have raised more serious and valid concerns, I still think that this is a good paper. The strength of the paper is very convincing and novel experimental results. Although the proposed model is not ideal, even with such a model the paper is in a good position to initiate discussions and further studies of ultrafast magnetism and optical control of spin textures in vdW materials.

Reviewer #3:

Remarks to the Author:

The topic of this work, i.e., all-optical topological switching, is very interesting, but I am not so convinced about the theoretical model used in this work. The current version is suitable for the publication of this work in Nature Communications. Here are my comments:

1. On page 3, lines 53-56, the authors cite many references for different topological spin textures reported in 2D magnets. All of them are experimental ones. Although the authors apply a multiscale approach (DFT + spin model) for the theory part, none of the previous theoretical work (see below) has been cited...

Fe₃GeTe₂

K. Huang et al., Nano Lett. 22, 3349 (2022),
<https://pubs.acs.org/doi/10.1021/acs.nanolett.2c00564>

D. Li et al., Nano Lett. 22, 7706 (2022),
<https://pubs.acs.org/doi/full/10.1021/acs.nanolett.2c03287>

Cr₂Ge₂Te₆

Chao-Kai Li et al., Phys. Rev. Research 3, L012026 (2021),
<https://journals.aps.org/prresearch/abstract/10.1103/PhysRevResearch.3.L012026>

2. As far as I know, the first experimental demonstration of all-optical topological switching was reported by "F. Büttner et al., Nat. Mater. 20, 30 (2021)". The author should cite this pioneering work.

3. The recent paper [NPJ Computational Materials, 169 (2022)] published by some of the authors they have already demonstrated the optical control of topological magnetism switching (i.e., meron and antimeron). The main difference is that they studied different materials. So, for me, the current paper is only an extension of their previous work, as I pointed out in my previous report. Can the authors comment on this explicitly in the manuscript?

4. The authors do not actually change magnetic interaction parameters under the laser, but rather the ultrashort pulses change the spin dynamics under different fluences. To describe the ultrafast magnetism accurately under a laser, we need a theoretical framework based on the "TDDFT+spin model". Can the authors comment on the difference between the 2TM model and with "TDDFT+spin model"?

NCOMMS-22-38270-A

REVIEWER COMMENTS

Reviewer #1 (Remarks to the Author):

The authors have addressed all my comments. And I would like to recommend the publication in NC.

Response 1: *We thank the Reviewer for accepting our manuscript for publication in Nature Communications.*

Reviewer #2 (Remarks to the Author):

The authors have adequately addressed my concerns in the revised version of the manuscript. Although the other two Referees have raised more serious and valid concerns, I still think that this is a good paper. The strength of the paper is very convincing and novel experimental results. Although the proposed model is not ideal, even with such a model the paper is in a good position to initiate discussions and further studies of ultrafast magnetism and optical control of spin textures in vdW materials.

Response 2: *We thank the Reviewer for the additional comments and for accepting our manuscript for publication in Nature Communications.*

Reviewer #3 (Remarks to the Author):

The topic of this work, i.e., all-optical topological switching, is very interesting, but I am not so convinced about the theoretical model used in this work. The current version is suitable for the publication of this work in Nature Communications. Here are my comments:

Response 3: *We thank the Reviewer for the additional comments and for accepting our manuscript for publication in Nature Communications. We have included further discussions and references following the Reviewer's comments. They are included in the updated version of the manuscript.*

1. On page 3, lines 53-56, the authors cite many references for different topological spin textures reported in 2D magnets. All of them are experimental ones. Although the authors apply a multiscale approach (DFT + spin model) for the theory part, none of the previous theoretical work (see below) has been cited...

Fe₃GeTe₂

K. Huang et al., Nano Lett. 22, 3349

(2022), <https://pubs.acs.org/doi/10.1021/acs.nanolett.2c00564>

D. Li et al., Nano Lett. 22, 7706

(2022), <https://pubs.acs.org/doi/full/10.1021/acs.nanolett.2c03287>

Cr₂Ge₂Te₆

Chao-Kai Li et al., Phys. Rev. Research 3, L012026

(2021), <https://journals.aps.org/prresearch/abstract/10.1103/PhysRevResearch.3.L012026>

Response 4: *We thank the Reviewer for suggesting the references which have been included in the manuscript.*

2. As far as I know, the first experimental demonstration of all-optical topological switching was reported by "F. Büttner et al., Nat. Mater. 20, 30 (2021)". The author should cite this pioneering work.

Response 5: We thank the Reviewer for mentioning this reference. It is important to remark that the switching behaviour reported by F. Büttner et al. is not reversible, as it is in our case. That is, they seem to show only one-way switching (either from a uniform magnetisation state, or stripe domains, into skyrmions, but not vice-versa). In our manuscript however we observed that we can induce the formation of skyrmions from stripes, and reverse the process obtained the original state used, which can be either topological spin textures or stripes. In this case, our manuscript, to the best of our knowledge, is the first demonstration of a reversible topological spin-switch on a 2D vdW magnet.

We have included additional comments regarding this process and comparison with the results by F. Büttner et al. at page 13, lines 251-258 in the updated version of the manuscript.

3. The recent paper [NPJ Computational Materials, 169 (2022)] published by some of the authors they have already demonstrated the optical control of topological magnetism switching (i.e., meron and antimeron). The main difference is that they studied different materials. So, for me, the current paper is only an extension of their previous work, as I pointed out in my previous report. Can the authors comment on this explicitly in the manuscript?

Response 4: We thank the Reviewer for the comments and to point out this reference. In the mentioned paper [Strungaru et al. npj Comput. Mater. 2022, 169], only the laser-driven formation of spin textures (e.g., merons or anti-merons) from a homogeneous magnetic state was reported. The results in Strungaru et al. do not include: 1) any switching mechanism between different spin states (e.g., stripes into skyrmions, and vice-versa); 2) formation of different spin textures (e.g., skyrmions, anti-skyrmions, skyrmionium); and, finally, 3) phase diagrams for potential selection of spin states for spin-toggle switch processes, as it is reported in the present manuscript under consideration. These differences are key and provide a fresh view of topological spin-switching phenomena present in 2D magnets.

We have included additional discussions in the manuscript highlighting the differences both manuscripts in page 14, lines 275-279.

4. The authors do not actually change magnetic interaction parameters under the laser, but rather the ultrashort pulses change the spin dynamics under different fluences. To describe the ultrafast magnetism accurately under a laser, we need a theoretical framework based on the "TDDFT+spin model". Can the authors comment on the difference between the 2TM model and with "TDDFT+spin model"?

Response 5: We thank the Reviewer for the comments. The theoretical framework used to simulate the time-evolution of the magnetisation dynamics is based on atomistic spin dynamic simulations using the stochastic Landau-Lifshitz-Gilbert (LLG) extended to the 2-temperature model (2TM). Such approach allows long-time simulations as those included in our manuscript (>8 ns), which are unpractical in TDDFT. Moreover, most of the topological phenomena occurred deeply into the thermal equilibration beyond 1 ns at large areas (e.g., 250 nm x 250 nm) not accessible using TDDFT methods. The close agreement between experimental results and simulations supports our choice of using LLG-2TM in our work.

We have included additional comments in the manuscript regarding the long-time scale required to model the laser processes and comparison with TDDFT methods in pages 15-16, lines 303-314.